# The role of PNI to predict survival in advanced hepatocellular carcinoma treated with Sorafenib

Francesco Caputo[1], Vincenzo Dadduzio[2], Francesco Tovoli[3], Giulia Bertolini[4], Giuseppe Cabibbo[5], Krisida Cerma[1], Caterina Vivaldi[6], Luca Faloppi[7], Mario Domenico Rizzato[2,8], Fabio Piscaglia[3], Ciro Celsa[5], Lorenzo Fornaro[6], Giorgia Marisi[4], Fabio Conti[9], Nicola Silvestris[10], Marianna Silletta[11], Sara Lonardi[2], Alessandro Granito[3], Caterina Stornello[12], Valentina Massa[6], Giorgio Astara[13], Sabina Delcuratolo[10], Stefano Cascinu[14], Mario Scartozzi[13], Andrea Casadei-Gardini[1]*

1 Division of Oncology, Department of Oncology and Hematology, University of Modena and Reggio Emilia, Modena, Italy, 2 Medical Oncology 1, Veneto Institute of Oncology IOV-IRCCS, Padua, Italy, 3 Azienda Ospedaliera Universitaria S.Orsola-Malpighi Bologna, Bologna, Italy, 4 Medical Oncology Unit IRCSS-IRST Meldola, Meldola, Italy, 5 Section of Gastroenterology & Hepatology, PROMISE, University of Palermo, Palermo, Italy, 6 Department of Oncology, University of Pisa, Pisa, Italy, 7 Medical Oncology Unit, Macerata General Hospital, Macerata, Italy, 8 Department of Surgery, Oncology and Gastroenterology, University of Padua, Padua, Italy, 9 Department of Internal Medicine, Degli Infermi Hospital, Faenza, Italy, 10 Medical Oncology Unit, IRCCS Giovanni Paolo II Cancer Center, Bari, Italy, 11 Medical Oncology Department, Campus Biomedico, University of Rome, Rome, Italy, 12 Digestive and Liver Disease Unit, S. Andrea Hospital, Rome, Italy, 13 Department of Medical Oncology, University of Cagliari, Cagliari, Italy, 14 Department of Medical Oncology, IRCCS San Raffaele Scientific Institute, Milan, Italy

* casadeigardini@gmail.com

**Data Availability Statement:** All relevant data are within the paper and its Supporting Information files.

## Abstract

### Background and aims

The present study aims to investigate the role of the prognostic nutritional index (PNI) on survival in patients with advanced hepatocellular carcinoma (HCC) treated with sorafenib.

### Methods

This multicentric study included a training cohort of 194 HCC patients and three external validation cohorts of 129, 76 and 265 HCC patients treated with Sorafenib, respectively. The PNI was calculated as follows: 10 × serum albumin (g/dL) + 0.005 × total lymphocyte count (per mm$^3$). Univariate and multivariate analyses were performed to investigate the association between the covariates and the overall survival (OS).

### Results

A PNI cut-off value of 31.3 was established using the ROC analysis. In the training cohort, the median OS was 14.8 months (95% CI 12–76.3) and 6.8 months (95% CI 2.7–24.6) for patients with a high (>31.3) and low (<31.3) PNI, respectively. At both the univariate and the multivariate analysis, low PNI value (p = 0.0004), a 1-unit increase of aspartate aminotransferase (p = 0.0001), and age > 70 years (p< 0.0038) were independent prognostic factors for OS. By performing the same multivariate analysis of the training cohort, the PNI <31.3

**Funding:** The author(s) received no specific funding for this work.

**Competing interests:** The authors have declared that no competing interests exist.

**Abbreviations:** HCC, Hepatocellular carcinoma; BCLC, Barcelona Clinic Liver Cancer; BMI, Body Mass Index; MVI, macroscopic vascular invasion; ALBI, Albumin-Bilirubin; AFP, Alpha-fetoprotein; LDH, Lactate Dehydrogenase; NLR, neutrophil-to-lymphocyte ratio; SII, immune-inflammation index; AE, adverse events; PNI, prognostic nutritional index; ECOG, Eastern Cooperative Oncology Group; ALT, alanine aminotransferase; AST, aspartate aminotransferase; mRECIST, modified Response Evaluation Criteria in Solid Tumors; OS, Overall Survival; HBV, hepatitis B virus; HCV, hepatitis C virus; mPFS, median progression-free survival; PD, progression disease; HRs, Hazard ratios; DFS, disease-free survival; CLIP, Cancer of the Liver Italian Program; TACE, transarterial chemoembolization; G-GT, gamma-glutamyl transpeptidase.

versus >31.3 was found to be an independent prognostic factor for predicting OS in all the three validation cohorts.

## Conclusions

PNI represents a prognostic tool in advanced HCC treated with first-line Sorafenib. It is readily available and low-cost, and it could be implemented in clinical practice in patients with HCC.

## 1. Introduction

Hepatocellular carcinoma (HCC) is the most common primary malignancy of the liver [1]. It represents the fifth most common cancer worldwide and the second cause of cancer mortality [2]. Sorafenib, an oral multikinase inhibitor (VEGFR- 1/2/3, PDGFR, Flt3, c-Kit, and Raf kinases), has been considered the standard of care for patients with advanced unresectable HCC since 2007 [3,4]. In the literature, several clinical and biochemical factors have been described as predictive or prognostic markers in patients with HCC treated with Sorafenib, such as etiology [5–7], Child-Pugh status [8], Barcelona Clinic Liver Cancer (BCLC) [5], medical drugs [9,10], Body Mass Index (BMI) [11], macroscopic vascular invasion (MVI) [5], Aspartate aminotransferase (AST) [12], Albumin-Bilirubin (ALBI) grade [13], Alpha-fetoprotein (AFP) [5,14,15], Lactate Dehydrogenase (LDH) [15,16], Neutrophil-to-lymphocyte ratio (NLR) [5,16,17], Immune-inflammation index (SII) [17,18], and drug-related adverse events (AE) [19,20]. In addition, the pattern of progression [21] and the reason for Sorafenib discontinuation [22] have been reported to have a significant correlation with survival. Furthermore, biological parameters, as serum and plasma proteins [23], genetic markers [24–26], micro-RNAs [27], and tissue biomarkers [28] have been investigated. At the moment other potential new biomarkers are under exploration, as the so-called OMICS revolution, Radiomics, and Liquid Biopsy [29].None, of these factors has been validated [30] and they are not usually used in clinical practice to select patients in making clinical decisions.

In this context, the prognostic nutritional index (PNI), is a multiparametric indicator based on serum albumin and peripheral lymphocyte count [31], has shown to reflect both the immune-inflammatory and nutritional status of patients [32], even if in HCC it particularly reflects the liver disfunction underlying this cancer. Interestingly, the PNI has demonstrated to correlate with survival outcomes of patients with several gastrointestinal cancers [33,34], including HCC [35–37] but not in advanced stage. [38].

In the present study, we have investigated the impact of the PNI index on survival outcomes in four independent cohorts of advanced HCC treated with sorafenib.

## 2. Materials and methods

This multicentric Italian study was conducted on a training cohort of 194 HCC patients consecutively treated at Istituto Scientifico Romagnolo per lo Studio e la Cura dei Tumori from 2007 to 2015. Three validation cohorts of HCC patients were consecutively recruited by the University of Bologna for the first cohort, a multicentric prospective study (INNOVATE study) [39] for the second cohort and from the University of Palermo and IOV Veneto of Padua for the third cohort.

Patients with histologically or radiologically (according to the American Association for the Study of Liver Diseases 2005 guidelines) proven advanced- or intermediate-stage (refractory or unsuitable for loco-regional therapies) HCC treated with sorafenib in real life were eligible for our analysis. Patients who had received previous systemic therapies were excluded. All patients received sorafenib according to standard schedule (400 mg bid continuously); dose reduction was applied as clinically indicated. Follow-up consisted of a CT/MRI scan every 8 weeks or as clinically indicated. Tumor response was evaluated by modified Response Evaluation Criteria in Solid Tumors (mRECIST) [43]. Treatment with sorafenib was continued until disease progression, unacceptable toxicity or death.

The study protocol was reviewed and approved by the local Ethics Committee (CEIIAV: comitato etico IRST IRCCS AVR and CE-AVEC: comitato etico Bologna). Study number IRST B041 protocol number 5482/v.1 intern code: L3P1192. Study number Bologna 098/2014/U/Oss. All patients provided written informed consent.

## 2.1 Statistical analysis

This analysis aimed to examine the association between baseline PNI index and Overall Survival (OS) in patients with HCC treated with sorafenib.

Information on neutrophil and albumin from hematologic blood tests carried out at baseline (the day before the start of treatment) was collected.

The PNI was calculated as follows: $10 \times$ serum albumin concentration (g/dL) $+ 0.005 \times$ peripheral lymphocyte count (number/mm$^2$) [34]. The cut-off point of the PNI was determined to be 31.3 by ROC analysis.

Categorical variables were compared with Fisher's exact test.

OS was defined as the time interval from the first day of treatment to the day of death or last follow-up visit. OS was estimated by the Kaplan-Meier method and curves were compared by the log-rank test. Unadjusted and adjusted hazard ratios (HRs) by baseline characteristics (AST, Age, and Sex) were calculated using the Cox proportional hazards model. The discrimination ability of the final model was assessed with Harrell's concordance index (C-index).

MedCalc package (MedCalc® version 16.8.4) was used for statistical analysis.

## 3. Results

Among the 194 Sorafenib-treated HCC patients of the training group, 168 (86.6%) were males and 26 (13.4%) were females, with a median age of 70 years (range 25–87). Seventy-nine patients (40.7%) had an ECOG 0. The underlying etiology of liver disease was hepatitis B virus (HBV) in 45 patients (25.2%), hepatitis C virus (HCV) in 107 patients (55.1%), alcohol in 8 patients (4.1%), metabolic syndrome in 16 patients (8.2%), others in 18 patients (9.3%). The Child-Pugh Class was A in 157 patients (80.9%) and B in 30 patients (15.6%). Other baseline clinicopathologic and laboratory characteristics are summarized in **Table 1**.

A total of 20 (10.3%) patients was categorized as the PNI-low group, while the remaining 174 (89.7%) patients as the PNI-high group.

Eighty-one (41.7%) patients had early AE, while 57 (29.4%) patients had late AE. The most frequent drug-related AEs were dermatologic toxicity and diarrhea.

The median progression-free survival (mPFS) was 3,8 months (95% CI, 3.1–6.4) and the median overall survival (mOS) was 12.4 (95% CI, 11.3–24.6).

### 3.1 Prognostic value of the PNI in the training cohort

At the univariate analysis for OS high PNI was associated with longer mOS (14.8 vs 6.8 months, HR 5.26; 95% CI,2.49–11.10; p<0.0001) **(Fig 1)**. In addition, normal level of AST (<1

**Table 1. Baseline characteristics of the training cohort.**

| Parameters | N (%) |
|---|---|
| **Age, years (median, range)** | 70 (25–87) |
| **Gender** | |
| Female | 26 (13.4%) |
| Male | 168 (86.6%) |
| **ECOG PS** | |
| 0–1 | 87 (44.8%) |
| ≥2 | 79 (40.7%) |
| Unknown | 28 (14.5%) |
| **Etiology** | |
| HCV | 107 (55.2%) |
| HBV | 45 (23.2%) |
| Metabolic syndrome | 16 (8.3%) |
| Alcohol | 8 (4.1%) |
| Others | 18 (9.2%) |
| **Child-Pugh** | |
| A | 157 (80.9%) |
| B | 30 (15.5%) |
| Unknown | 7 (3.6%) |
| **BCLC Stage** | |
| B | 39 (20.1%) |
| C | 127 (65.4%) |
| Unknown | 28 (14.5%) |
| **"Local" Treatment** | |
| Liver transplantation | 4 (2.0%) |
| Radiofrequency | 29 (14.9%) |
| TACE | 73 (37.6%) |
| **PNI** | |
| Low-group (<31.3) | 20 (10.3%) |
| High-group (≥31.3) | 174 (89.7%) |
| **ALBI grade** | |
| 0 | 0 |
| 1 | 188 (98.0) |
| 2 | 4 (2.0) |
| **Laboratory tests (median, range)** | |
| Neutrophils, cells/μl | 3540 (290–11490) |
| Lymphocytes, cells/μl | 1300 (210–3560) |
| Platelets, cells/μl | 137 (44–462) |
| Albumin, gr/dl | 3.6 (2.6–5.0) |
| Alkaline Phosphatase, IU/L | 144.5 (69–698) |
| G-GT, IU/L | 162 (45–1102) |
| AST, IU/L | 56 (13–177) |
| ALT, IU/L | 46 (10–218) |
| Bilirubin, gr/dl | 0.91 (0.3–3.86) |
| AFP, ng/ml | 23 (0.8–50000) |

**Abbreviations.** ECOG PS, eastern cooperative oncology group performance status; HCV, hepatitis C virus; HBV, hepatitis B virus; BCLC stage, Barcelona clinic liver center staging; Child-Pugh, Child-Turcotte-Pugh score; TACE, transarterial chemoembolization; PNI, prognostic nutritional index; G-GT, gamma-glutamyl transpeptidase; ALT, alanine aminotransaminase; AST, aspartate aminotransaminase; AFP, alpha-fetoprotein.

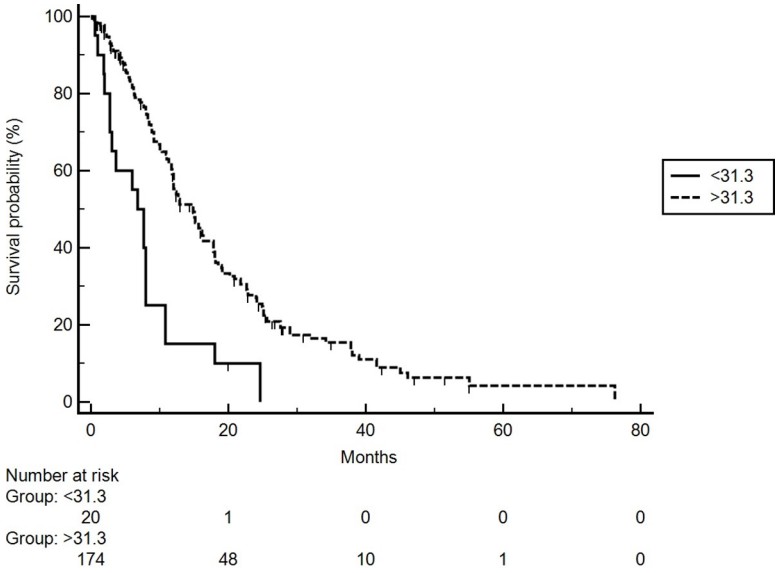

**Fig 1. OS according to PNI (high- vs low-group) in the training cohort.**

ULN) was correlated with better prognosis (HR 0.45; 95% CI, 0.30–0.68; p = 0.0002). No other correlations were found, particularly neither ALBI grade nor Child-Pugh status nor BCLC class were associated with prognosis, maybe due to the small number of patients in the worst classes. Nevertheless, we observed a trend towards a worse prognosis for these patients (ALBI grade: HR 2.47; 95% CI: 0.41–14.8; p = 0.1085; Child-Pugh status: HR 1.45; 95% CI: 0.90–2.32; p = 0.1312; BCLC class: HR 1,37; 95% CI: 0.92–2.03; p = 0.1132).

Following adjustment for clinical covariates positive in univariate analysis, multivariate analysis confirmed PNI-low (HR 2.98; 95% CI: 1.63–5.47; p = 0.0004), a normal level of AST (HR 0.33; 95% CI: 0.2–0.5; p = 0.0001), and age >70 years (HR 0.54; 95% CI: 0.36–0.82; p<0.0038) as independent prognostic factors for OS (**Table 2**).

Patients with a low PNI index showed a higher percentage of progression disease (PD) at the first CT re-evaluation respect to patients with a high PNI index (40% vs. 15% respectively, p = 0.04).

Next, we evaluated PNI index modifications during the early course of treatment. We estimated OS after stratifying patients into 3 groups according to PNI levels at baseline and after one month. The first group included patients with low (<31.3)-low (<31.3) levels of PNI index (11 patients), the second included those with high (>31.3)-low (<31.3) PNI index (36 patients) and the third included those with high (>31.3)-high (>31.3) PNI index (72 patients). No patients were classified as PNI low (<31,3)-PNI high (>31,3). Patients in the first group had a median OS of 7.7 months compared to 12.4 months for those in the second group and 15.1 months for those in the third group (p = 0.0029) (**Fig 2**).

### 3.2 Validation cohorts

A total of three external validation cohorts were considered for the analysis. 129 patients diagnosed with HCC were taken from the Bologna center database and made up the first validation cohort. Baseline clinical and laboratory characteristics are summarized in **Table 3**. Globally, 104 (80.6%) patients were categorized as the PNI-high group, while the remaining 25 (19.4%) patients as the PNI-low group. Maintaining the same observations of the training cohort, patients with PNI-low had a mOS of 4.0 months, whereas patients with a PNI-high had a mOS

**Table 2. Univariate and multivariate analysis in the training cohort.**

| Covariate | Univariate analysis | | | Multivariate analysis | | |
|---|---|---|---|---|---|---|
| | HR | 95%CI | p-value | HR | 95%CI | p-value |
| **Sex (female vs male)** | 1.29 | 0.77–2.16 | 0.3164 | - | - | - |
| **Age (>70 vs <70)** | 0.98 | 0.71–1.35 | 0,9295 | 0.54 | 0.36–0.82 | **0,0038** |
| **ECOG PS (>0 vs 0)** | 1.34 | 0.95–1.91 | 0.0985 | | | |
| **ALBI grade (2 vs 1)** | 2.47 | 0.41–14.8 | 0.1085 | | | |
| **Etiology** | | | | - | - | - |
| HCV | 1.00 | | | | | |
| HBV | 1.04 | 0.71–1.51 | | | | |
| NASH | 0.93 | 0.53–1.63 | | | | |
| Alcool | 0.86 | 0.40–1.83 | | | | |
| Others | 1.41 | 0.75–2.65 | 0.7398 | | | |
| **PNI (<31.3 vs >31.3)** | 0,19 | 0,09–0,4 | **<0.0001** | 2.98 | 1.63–5.47 | **0,0004** |
| **AST (>NV vs NV)** | 2.21 | 1.46–3.35 | **0.0002** | 0.33 | 0.20–0.57 | **0,0001** |
| **NLR (>3 vs <3)** | 1,03 | 0.73–1,44 | 0.8629 | - | - | - |
| **BCLC (C vs B)** | 1.37 | 0.92–2.03 | 0,1132 | - | - | - |
| **Bilirubin (>NV vs NV)** | 1,29 | 0,88–1,90 | 0,1873 | | | |
| **ALT (>NV vs NV)** | 1.22 | 0,84–1,87 | 0,2955 | | | |
| **Child-Pugh (B vs A)** | 1.45 | 0.90–2.32 | 0,1213 | | | |
| **AFP (>400 vs <400)** | 1.27 | 0,87–1,87 | 0,2084 | | | |
| **Reason for sorafenib discontinuation** | | | | | | |
| **Tumor progression vs. AE** Clinical decompensation Clinical decompensation | 1.27 | 0.91–1.45 | **0.163** | | | |

**Abbreviations.** PNI, prognostic nutritional index; AST, aspartate aminotransaminase; NV, normal value; NLR, neutrophil-to-lymphocyte ratio; BCLC stage, Barcelona clinic liver center staging; ECOG PS, eastern cooperative oncology group performance status; ALT, alanine aminotransaminase; Child-Pugh, Child-Turcotte-Pugh score; AFP, alpha-fetoprotein; AE, adverse event.

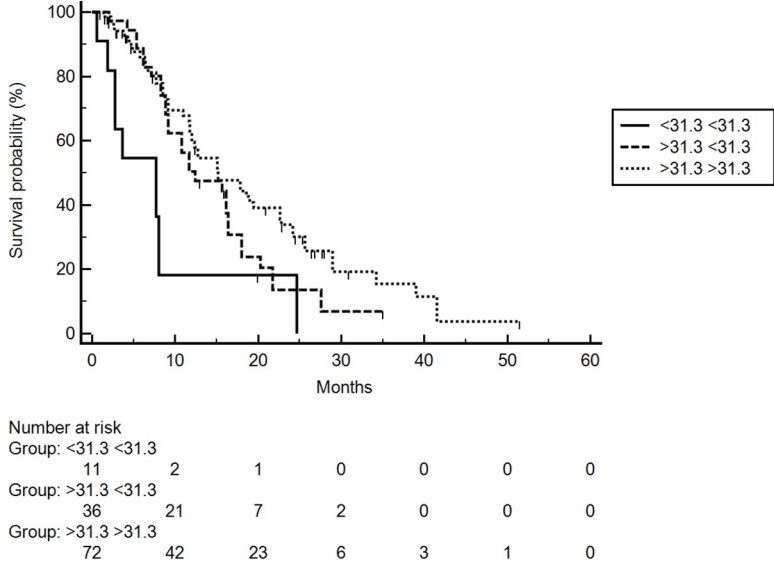

**Fig 2. OS according to PNI value changes after a month of treatment with Sorafenib in the training cohort.**

**Table 3. Baseline characteristics of the first, second- and third-validation cohort.**

| Parameters | N (%) | | |
|---|---|---|---|
| | **First validation cohort** | **Second validation cohort** | **Third validation cohort** |
| **Age, years (median, range)** | 67 (37–85) | 67 (24–84) | 66.8 (24–85) |
| **Gender** | | | |
| Female | 17 (13.2%) | 12 (15.8%) | 52 (19.6%) |
| Male | 112 (86.8%) | 64 (84.2%) | 213 (80.4%) |
| **ECOG PS** | | | |
| 0–1 | 94 (72.8%) | 75 (98.7%) | 253 (94.5%) |
| ≥2 | 35 (27.2%) | 1 (1.3%) | 11 (4.1%) |
| Unknown | 0 (0%) | 0 (0%) | 1 (0.4%) |
| **Etiology** | | | |
| HCV | 68 (52.7%) | 26 (34.2%) | 104 (39.2%) |
| HBV | 20 (15.5%) | 8 (10.5%) | 34 (12.8%) |
| Metabolic syndrome | 41 (31.8%) | 6 (7.9%) | 17 (6.4%) |
| Alcol | 0 (0%) | 6 (7.9%) | 37 (13.9%) |
| Other | 0 (0%) | 30 (39.5%) | 73 (27.7%) |
| **Child-Pugh** | | | |
| A | 121 (93.8%) | 65 (85.5%) | 238 (89.8) |
| B | 8 (6.2%) | 11 (14.5%) | 27 (10.2) |
| **BCLC Stage** | | | |
| B | 27 (21.0%) | 20 (26.3%) | 53 (20.0%) |
| C | 102 (79.0%) | 56 (73.7%) | 212 (80.0%) |
| **"Local" Treatment** | | | |
| Liver transplantation | 45 (34.8%) | NA | NA |
| Radiofrequency | 23 (17.8%) | NA | NA |
| TACE | 66 (51.1%) | NA | NA |
| **PNI** | | | |
| Low-group (<31.3) | 25 (19.4%) | 15 (19.7%) | 34 (12.8%) |
| High-group (≥31.3) | 104 (80.6%) | 61 (80.3%) | 231 (87.2%) |
| **Laboratory tests (median, range)** | | | |
| Neutrophils, cells/μl | 3590 (750–10200) | 4100 (730–600000) | 4180 (1100–12740) |
| Lymphocytes, cells/μl | 1130 (250–4800) | 1505 (590–6850) | 1300 (102–4250) |
| Platelets, cells/μl | 138 (26–400) | NA | 123 (13–483) |
| Albumin, gr/dl | 3.6 (2.7–5.0) | NA | 3.7 (2.6–5.3) |
| Alkaline Phosphatase, IU/L | 191 (44–1231) | NA | NA |
| G-GT, IU/L | 108 (17–1043) | NA | NA |
| AST, IU/L | 53 (9–334) | 57 (13–238) | 51 (19–300) |
| ALT, IU/L | 41 (5–300) | 43 (6–348) | NA |
| Bilirubin, gr/dl | 0.94 (0.2–3.04) | 0.81 (0.21–3.15) | NA |
| AFP, ng/ml | 30 (1.0–60500) | 44.1 (0.8–>50000) | 72.7 (1.1–457300) |

**Abbreviations.** ECOG PS, eastern cooperative oncology group performance status; HCV, hepatitis C virus; HBV, hepatitis B virus; BCLC stage, Barcelona clinic liver center staging; Child-Pugh, Child-Turcotte-Pugh score; TACE, transarterial chemoembolization; NA, not available; PNI, prognostic nutritional index; G-GT, gamma-glutamyl transpeptidase; ALT, alanine aminotransaminase; AST, aspartate aminotransaminase; AFP, alpha-fetoprotein;

of 10.9 months (HR 0.03; 95% CI 0.01–0.08, p<0.0001). By performing the same multivariate analysis of the training cohort, PNI-low was found to be an independent prognostic factor for OS (HR 6.53; 95% CI 3.79–11.25, p<0.0001) (Fig 3A). The model had a C-index of 0.78.

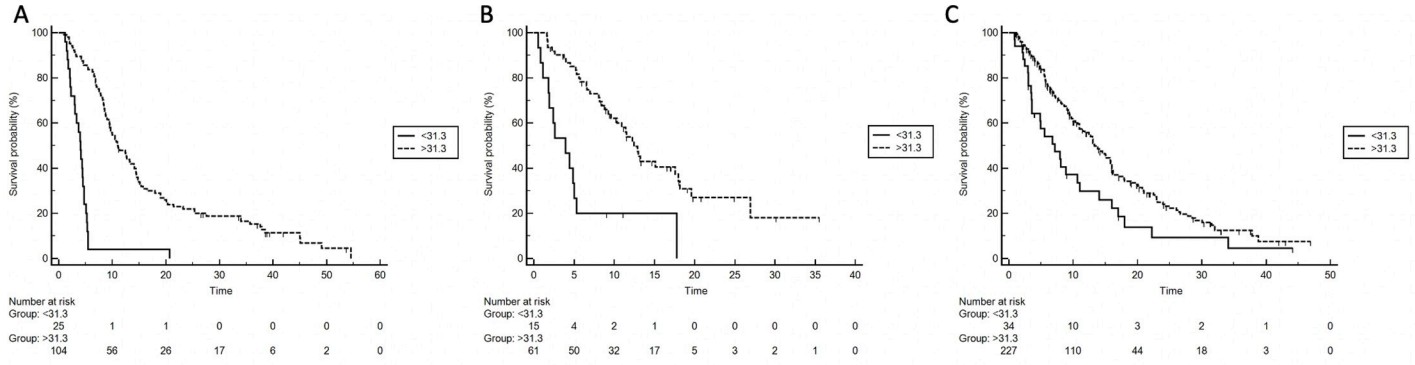

**Fig 3.** OS according to PNI in the first validation cohort (A); second validation cohort (B) and third validation cohort (C).

The participants in a second validation cohort were 76 patients collected in the Innovate STUDY database [42]. Baseline clinical and laboratory characteristics are summarized in **Table 3**.

Globally, 61 (80.3%) patients were categorized as the PNI-high group, while the remaining 15 (19.7%) patients as the PNI-low group. Patients with PNI-low had a mOS of 3.9 months, whereas patients with a PNI-high had a mOS of 12.4 months (HR 0.1, 95% CI 0.04–0.31, p<0.0001) (**Fig 3B**). By performing the same multivariate analysis of the training cohort, PNI-low was found to be an independent prognostic factor for OS (HR 2.98, 95% CI 1.25–7.08; p = 0.0135). The model had a C-index of 0.73.

The participants in a third validation cohort were 265 patients taken from Palermo and Padua centers. Baseline clinical and laboratory characteristics are summarized in **Table 3**.

Overall, 231 (87.2%) patients were categorized as the PNI-high group, while the remaining 34 (12.8%) patients as the PNI-low group. Patients with PNI-low had a mOS of 7.1 months, whereas patients with a PNI-high had a mOS of 13.3 months (HR 0.47, 95% CI 0.28–0.78, p = 0.0037). By performing the same multivariate analysis of the training cohort, PNI-low was found to be an independent prognostic factor for OS (HR 1.94, 95% CI 1.22–4.58; p = 0.001) (**Fig 3C**). The model had a C-index of 0.77.

Multivariate analyses of the three validation cohorts are summarized in **Table 4**.

## 4. Discussion

In this large retrospective study totalling a number of 660 patients, including three independent validation cohorts of subjects, we demonstrated that the PNI is an independent predictor of survival.

**Table 4. Multivariate analysis in the first, second and third validation cohort.**

| Covariate | Multivariate analysis | | | | | | | | |
|---|---|---|---|---|---|---|---|---|---|
| | First validation cohort | | | Second validation cohort | | | Third validation cohort | | |
| | HR | 95%CI | p-value | HR | 95%CI | p-value | HR | 95%CI | p-value |
| PNI <31.3 | 6.53 | 3.79–11.25 | <**0.0001** | 2.98 | 1.25–7.08 | **0.0135** | 1.94 | 1.22–4.58 | **0.0012** |
| AST | 0.81 | 0.54–1.22 | 0.3326 | 0.91 | 0.44–1.89 | 0.8099 | 1.00 | 0.99–1.01 | 0.2494 |
| Age | 0.77 | 0.51–1.16 | 0.2189 | 1.43 | 0.73–2.80 | 0.2873 | 0.55 | 0.25–1.22 | 0.1435 |
| Sex | 1.13 | 0.63–2.03 | 0.6774 | 2.39 | 0.98–5.78 | 0.0529 | 0.77 | 0.33–1.80 | 0.5529 |

**Abbreviations**. PNI, prognostic nutritional index; AST, aspartate aminotransaminase.

PNI index is composed of only two parameters. This simplification of the index makes it more available and simpler for daily clinical practice respect others index evaluated by our and others groups [5,8,35–38].

The results of our analysis are in line with previous studies evaluating the prognostic role of this index. Particularly, Chan et al reported that the PNI predicts tumor recurrence in early-stage HCC after surgical resection [39]; it is also associated with survival of HCC patients after loco-regional or systemic therapy, as reported by Pinato et al [40]. Furthermore, a meta-analysis of eleven studies also proved that a low PNI is a poor prognostic factor for OS and disease-free survival (DFS) in HCC, whereas a high PNI is a favourable prognostic factor and is associated with better clinical predictors, such as lower AFP, lower recurrence rates, smaller tumor size, and earlier TNM tumor stage [41].

In the Japanese experience of Hatanaka et al [42], the PNI was a significant factor associated with the duration of Sorafenib therapy and the OS among pre-treatment factors in a cohort of patients treated with Sorafenib. No significant differences in terms of Sorafenib efficacy and serious adverse events rate were found between high- and low-PNI groups.

In keeping with the Japanese results, we showed the prognostic role of the PNI in a European population and we validated these results in three independent cohorts of patients with HCC treated with Sorafenib.

Several mechanisms could be put forward to explain how the PNI influences the survival of HCC patients, including those receiving sorafenib. First of all, the PNI, a combination of serum albumin and total lymphocyte count, reflects the link existing between immunity, inflammation, and nutrition in cancer, with their consequent potential prognostic implications. A low PNI may be caused by hypoalbuminemia and/or lymphocytopenia. In connection with the role of lymphocytopenia, it is well-known that T lymphocytes play an essential role in the immune and anti-cancer response and also in the biological behavior of HCC, such as initiation, proliferation, differentiation, and metastasis [43,44]. In previous studies on HCC, it has been reported that the presence of more abundant tumor-infiltrating effector T lymphocytes (TILs) is associated with better outcomes after surgical resection [45] while a reduced number appeared related to higher tumor recurrence rates after liver transplantation [46]. Similar results were observed also in advanced HCC, where the count of CD8+T cells in TILs was lower in patients with metastatic disease than in those without [47] and the ability of specific subsets of T cells in HCC was claimed to be able to predict extrahepatic metastasis and prognosis [48,49].

In connection with the role of hypoalbuminemia, albumin is a known prognostic factor in HCC, specifically included in staging systems like the Cancer of the Liver Italian Program (CLIP) score [50] or the ALBI grade [51] and in many other staging systems.

In this setting, albuminemia may be specifically influenced by three factors: 1) the liver dysfunction, usually related to the common underlying cirrhotic condition; 2) the nutritional status, including cancer cachexia; 3) the cancer-related inflammation. Furthermore, lower albumin levels were shown to be associated with increased risks of developing portal vein thrombosis in cirrhosis [52]. Whether similar effects may take place at a microvascular level leading to accelerated liver dysfunction remains a purely speculative hypothesis [53].

Interestingly, we were able to test and confirm in our training cohort not only that PNI level contributes to predicting survival when tested at baseline, but also when patients were subdivided into three groups according to the changes of PNI value over the first month of Sorafenib treatment, with the best OS observed in patients who could maintain high PNI levels in the first month of treatment.

Limitations of the present study was its retrospective nature and for this reason we didn't collected all clinical variables and the cohorts of the study were unbalance for clinical variables.

Other limitations of the study was the absence of a predefined standard cut-off value of the PNI. However, the cut-off obtained in our training group was validated in all the three external series, but it could strongly be validated by a prospective study. Another important limitation of our study is the absence of a control arm not receiving sorafenib, making not possible to evaluate the predictive role of the index.

In conclusion, we proposed the PNI as an easy-to-use prognostic factor in patients with HCC treated with Sorafenib, including nutritional status, inflammation, and immunity in a single marker. It is also readily available and low-cost, and for these reasons, it could be implemented in clinical practice and useful in trial design in patients with HCC.

## Supporting information

**S1 Data.**
(XLSX)

## Author Contributions

**Writing – original draft:** Francesco Caputo, Vincenzo Dadduzio, Francesco Tovoli, Giulia Bertolini, Giuseppe Cabibbo, Krisida Cerma, Caterina Vivaldi, Luca Faloppi, Mario Domenico Rizzato, Fabio Piscaglia, Ciro Celsa, Lorenzo Fornaro, Giorgia Marisi, Fabio Conti, Nicola Silvestris, Marianna Silletta, Sara Lonardi, Alessandro Granito, Caterina Stornello, Valentina Massa, Giorgio Astara, Sabina Delcuratolo, Stefano Cascinu, Mario Scartozzi, Andrea Casadei-Gardini.

**Writing – review & editing:** Francesco Caputo, Vincenzo Dadduzio, Francesco Tovoli, Giulia Bertolini, Giuseppe Cabibbo, Krisida Cerma, Caterina Vivaldi, Luca Faloppi, Mario Domenico Rizzato, Fabio Piscaglia, Ciro Celsa, Lorenzo Fornaro, Giorgia Marisi, Fabio Conti, Nicola Silvestris, Marianna Silletta, Sara Lonardi, Alessandro Granito, Caterina Stornello, Valentina Massa, Giorgio Astara, Sabina Delcuratolo, Stefano Cascinu, Mario Scartozzi, Andrea Casadei-Gardini.

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
