## [Decision Letter · Decision Letter 0]

8 Apr 2020

PONE-D-20-08153

THE ROLE OF PNI TO PREDICT SURVIVAL IN ADVANCED HEPATOCELLULAR CARCINOMA TREATED WITH SORAFENIB

PLOS ONE

Dear Dr. Casadei Gardini,

Thank you for submitting your manuscript to PLOS ONE. After careful consideration, we feel that it has merit but does not fully meet PLOS ONE’s publication criteria as it currently stands. Therefore, we invite you to submit a revised version of the manuscript that addresses the points raised by the reviewers. In addition, in your revised manuscript you should clarly highlight and discuss the novelty and differential value of your present study in view of previous works from your laboratory, such as Gastrointest Tumors, 2019;6:71-80  doi.org/10.1159/000501593 .

Specifically, you should argue, if this is the case, tha separation of the data into more than one article has not compromised the robustness of the statistical analysis of you current data.

We would appreciate receiving your revised manuscript by May 23 2020 11:59PM. To enhance the reproducibility of your results, we recommend that if applicable you deposit your laboratory protocols in protocols.io, where a protocol can be assigned its own identifier (DOI) such that it can be cited independently in the future. For instructions see: http://journals.plos.org/plosone/s/submission-guidelines#loc-laboratory-protocols

We look forward to receiving your revised manuscript.

Kind regards,

Matias A Avila, Ph.D.

Academic Editor

PLOS ONE

Journal Requirements:

Reviewers' comments:

Reviewer's Responses to Questions

**Comments to the Author**

1. Is the manuscript technically sound, and do the data support the conclusions?

Reviewer #1: Partly

Reviewer #2: Yes

2. Has the statistical analysis been performed appropriately and rigorously? 

Reviewer #1: Yes

Reviewer #2: Yes

3. Have the authors made all data underlying the findings in their manuscript fully available?

Reviewer #1: Yes

Reviewer #2: No

4. Is the manuscript presented in an intelligible fashion and written in standard English?

Reviewer #1: No

Reviewer #2: Yes

5. Review Comments to the Author

Reviewer #1: The Authors report the results of a retrospective study aimed to investigate the role of the prognostic nutritional index (PNI) on survival in patients with advanced hepatocellular carcinoma (HCC) treated with sorafenib. The topic is of interest considering the expanding landscape of treatment options for advanced HCC and the need to identify predictive and prognostic markers that can be of help in selecting patients to be treated with the available drugs. However, the retrospective nature of the study and the lack of a validated cutoff value for the PNI limit the robustness of the results.

The following comments should be addressed.

General comment

1. The English language should be revised

Major comments:

1. Introduction, page 4: Among the prognostic factors the Authors should mention the pattern of progression and the reason for sorafenib discontinuation as published by Reig M et al and Iavarone M et al, respectively (Hepatology 2013; Hepatology 2015). These factors should be included in the analyses

2. Results, Prognostic and predictive value of the PNI in training cohort: Did the Authors evaluate the predictive role of PNI or only the prognostic role? In the methods they stated that both have been assessed but no results are available for the predictive role. This aspect should be clarified and discussed in the discussion

3. Results, Validation cohorts: Baseline characteristics are partly different (e.g., ECOG PS, etiology) or missing (e.g., ALBI grade) in the validation cohorts. Did the Authors consider this factor in their analysis? This point should be also discussed in the discussion among the limitations of the study

4. References: The number of references from the same Authors should be reduced and references from other groups should be added

Minor comment:

1. References: Ref #1 should be replaced with a more recent paper

Reviewer #2: I read with interest the manuscript by Caputo et al. on the role of PNI to predict survival in advanced hepatocellular carcinoma treated with Sorafenib. Authors define the optimal cut-off of PNI independently associated with survival in a training cohort, and validate the finding in two external validation cohorts.

I have only the following minor points:

1) It was surprising to me seeing that, in the training cohort, neither ALBI grade nor Child-Pugh class nor BCLC class were associated with prognosis at the univariate analysis. It is probably due to relatively small number of patients in the worst classes. Do Authors have any other explication for this? In any case, I think this should be properly acknowledged and discussed.

2) In the Abstract, Results, the median OS of patients with low and high PNI has been inverted.

3) In The Methods, Authors state, among the eligibility criteria, there was a Child-Pugh liver class A. However, subsequently, it is clearly reported that also Child-Pugh B patients have been enrolled in the 3 cohorts.

6. PLOS authors have the option to publish the peer review history of their article (what does this mean?). If published, this will include your full peer review and any attached files.

Reviewer #1: No

Reviewer #2: No

---

## [Author Response · Author response to Decision Letter 0]

14 Apr 2020

Editor comment: In your revised manuscript you should clarly highlight and discuss the novelty and differential value of your present study in view of previous works from your laboratory, such as Gastrointest Tumors, 2019;6:71-80 doi.org/10.1159/000501593

REPLY: This study was different respect others our works because we evaluated here a simpler index than the previous. For example, RAPID index was composed about neutrophil, lymphocyte count, LDH and AFP, different PNI was com posed only by lymphocyte count and albumin. This simplification of the index makes it more available and simpler for daily clinical practice. Furthermore, we have increase the number and cohorts of patients than the previous studies. We have added this points in the discussion.

Reviewer #1: The Authors report the results of a retrospective study aimed to investigate the role of the prognostic nutritional index (PNI) on survival in patients with advanced hepatocellular carcinoma (HCC) treated with sorafenib. The topic is of interest considering the expanding landscape of treatment options for advanced HCC and the need to identify predictive and prognostic markers that can be of help in selecting patients to be treated with the available drugs. However, the retrospective nature of the study and the lack of a validated cutoff value for the PNI limit the robustness of the results.

The following comments should be addressed.

General comment

1. The English language should be revised

REPLY: We have improved the English language of the paper 

Major comments:

1. Introduction, page 4: Among the prognostic factors the Authors should mention the pattern of progression and the reason for sorafenib discontinuation as published by Reig M et al and Iavarone M et al, respectively (Hepatology 2013; Hepatology 2015). These factors should be included in the analyses

REPLY: WE have added in the introduction these two papers. We have performed the analysis about the reason for sorafenib discontinuation. Unfortunately, we didn’t have collected data about Pattern of progression for this reason the analyzes could not be performed. We have added this point in the limitation section.

2. Results, Prognostic and predictive value of the PNI in training cohort: Did the Authors evaluate the predictive role of PNI or only the prognostic role? In the methods they stated that both have been assessed but no results are available for the predictive role. This aspect should be clarified and discussed in the discussion

REPLY: We very thank the reviewer for this point. In absence of control arm is not possible to definite the prognostic role for this reason we have delete the sentence in the methods. We have added this point in the discussion.

3. Results, Validation cohorts: Baseline characteristics are partly different (e.g., ECOG PS, etiology) or missing (e.g., ALBI grade) in the validation cohorts. Did the Authors consider this factor in their analysis? This point should be also discussed in the discussion among the limitations of the study

REPLY: WE completely agree with the reviewer for this point. We have added it in the limitation section.

4. References: The number of references from the same Authors should be reduced and references from other groups should be added 

REPLY: We have reduced our references.

Minor comment:

1. References: Ref #1 should be replaced with a more recent paper

REPLY: We have changed the reference 1.

Reviewer #2: I read with interest the manuscript by Caputo et al. on the role of PNI to predict survival in advanced hepatocellular carcinoma treated with Sorafenib. Authors define the optimal cut-off of PNI independently associated with survival in a training cohort, and validate the finding in two external validation cohorts.

I have only the following minor points:

1) It was surprising to me seeing that, in the training cohort, neither ALBI grade nor Child-Pugh class nor BCLC class were associated with prognosis at the univariate analysis. It is probably due to relatively small number of patients in the worst classes. Do Authors have any other explication for this? In any case, I think this should be properly acknowledged and discussed.

REPLY: we observed a trend towards a worse prognosis for these patients. We agree with the reviewer that it was a consequence of a small number of patients. We have added this point in the result.

2) In the Abstract, Results, the median OS of patients with low and high PNI has been inverted.

REPLY: We very thank the reviewer for this point. 

3) In The Methods, Authors state, among the eligibility criteria, there was a Child-Pugh liver class A. However, subsequently, it is clearly reported that also Child-Pugh B patients have been enrolled in the 3 cohorts.

REPLY: Yes, we completely agree with the reviewer, we enrolled patients in real life. We have rewritten eligibility criteria of the study.

---

## [Editor Report · Decision Letter 1]

16 Apr 2020

THE ROLE OF PNI TO PREDICT SURVIVAL IN ADVANCED HEPATOCELLULAR CARCINOMA TREATED WITH SORAFENIB

PONE-D-20-08153R1

Dear Dr. Casadei Gardini,

We are pleased to inform you that your manuscript has been judged scientifically suitable for publication and will be formally accepted for publication once it complies with all outstanding technical requirements.

With kind regards,

Matias A Avila, Ph.D.

Academic Editor

PLOS ONE
---

## [Editor Report · Acceptance letter]

20 Apr 2020

PONE-D-20-08153R1 

THE ROLE OF PNI TO PREDICT SURVIVAL IN ADVANCED HEPATOCELLULAR CARCINOMA TREATED WITH SORAFENIB 

Dear Dr. Casadei-Gardini:

I am pleased to inform you that your manuscript has been deemed suitable for publication in PLOS ONE. Congratulations! Your manuscript is now with our production department. 

With kind regards,

on behalf of

Dr Matias A Avila 

Academic Editor

PLOS ONE